# Immunomodulation of IL-33 and IL-37 with Vitamin D in the Neointima of Coronary Artery: A Comparative Study between Balloon Angioplasty and Stent in Hyperlipidemic Microswine

**DOI:** 10.3390/ijms22168824

**Published:** 2021-08-17

**Authors:** Vikrant Rai, Devendra K. Agrawal

**Affiliations:** Department of Translational Research, Graduate College of Biomedical Sciences, Western University of Health Sciences, Pomona, CA 91766, USA; vrai@westernu.edu

**Keywords:** angioplasty, stent placement, IL-33, neointima, restenosis, coronary artery, vitamin D, immunomodulation

## Abstract

Inflammation is a major contributor to the development and progression of atherosclerosis. Interleukin (IL)-33 and IL-37, members of the IL-1 family, modulate inflammation, with IL-33 having a pro-inflammatory effect and IL-37 having anti-inflammatory properties. IL-37 is constitutively expressed at low levels but upregulated in inflammatory contexts. The aim of this study was to evaluate the effect of vitamin D on the expression of IL-33, IL-37, macrophages, and caspase-1 in the neointimal tissue of coronary artery in Yucatan microswine with vitamin D deficient, sufficient, and supplemented status. The intimal injury was induced by balloon angioplasty and stenting in the coronary artery, and tissues were harvested after 6 months. The expression of various proteins of interest was evaluated by immunostaining. Increased expression of IL-33 and IL-37 in the neointimal tissue of the vitamin D deficient, as compared to the sufficient and supplemented microswine, as revealed by histological evaluation and semi-quantitative analysis, suggested the immunomodulatory effect of vitamin D on the expression of IL-33 and IL-37. The minimal expression or absence of IL-33 and IL-37 expression in stented arteries is suggestive of an attenuated inflammatory response in stented arteries, compared to balloon angioplasty. The decreased IL-33 expression in the sufficient and supplemented microswine could be a potential mechanism for controlling the inflammatory process and neointima formation leading to attenuated luminal narrowing of the coronary artery. Overall, these results support supplementation of vitamin D to attenuate inflammation, neointima formation, and restenosis.

## 1. Introduction

Inflammation is known to play a vital role in the pathogenesis of atherosclerosis and coronary artery disease (CAD) [1,2,3]. Additionally, the roles of inflammation, arterial injury in neointima formation, and the development of restenosis after percutaneous intervention, including balloon angioplasty and stent placement, have been documented [4,5,6]. Immune cells play a crucial role in regulating inflammation in neointima and restenosis. The inflammatory response triggers a thrombotic and hyperplastic cascade of events, resulting in neointima formation and restenosis. The neointima formation and restenosis, after percutaneous coronary intervention, also depends on the inflammatory status before the intervention. Cells that participate in the innate immune response, including macrophages, T-lymphocytes, and dendritic cells (DCs), produce pro-inflammatory cytokines, including tumor necrosis factor (TNF)-α, interleukin (IL)-1, IL-6, and interferon (IFN)-γ. During adaptive immune responses, IFN-γ is produced, mainly by CD4+ T lymphocytes. There is an interplay between these and other cytokines that determines the balance between pro-inflammatory and anti-inflammatory cellular responses. It is this balance that, in part, determines the progression of atherosclerosis and CAD. While cytokines, such as IL-10 and transforming growth factor (TGF)-β, have been known for some time and are widely accepted as being anti-inflammatory, another anti-inflammatory cytokine IL-37 has only recently been discovered [3,4,5,6,7].

Vitamin D status has been implicated in contributing to the degree of the re-narrowing of atherosclerotic coronary arteries, following the coronary intervention, including balloon angioplasty and stenting; vitamin D deficiency exaggerates the neointimal hyperplasia lesions, and vitamin D supplementation results in a decreased neointima formation and luminal narrowing. The increased prevalence of inflammation and atherosclerosis associates with vitamin D deficiency and decreased luminal narrowing; additionally, neointimal formation associates with vitamin D supplementation [8,9,10]. Vitamin D is known to exert anti-inflammatory effects [11]; however, its effect on the expression of IL-33 and IL-37 post-percutaneous intervention had not been evaluated before this research. It is plausible that vitamin D may contribute to attenuate IL-33 and an increase in IL-37 expression within the neointimal lesions of coronary arteries as a protective mechanism preventing a re-narrowing, following coronary intervention. The inflammatory process of atherosclerosis elicits an immune response and anti-inflammatory mechanisms [12,13]; without the anti-inflammatory mechanism, vessels would quickly succumb to an unregulated inflammatory response. The expression of IL-37 is increased during inflammatory conditions and suppresses the inflammatory response [14]. This response of IL-37 to inflammatory processes, such as atherosclerosis, leads to the idea that it functions as a negative feedback mechanism, to diminish the atherogenic effects of excessive inflammation [7,15,16].

This study aimed to investigate the effect of vitamin D status on the expression of IL-33 and IL-37, the macrophages population, and to correlate the expression of IL-33 and IL-37 with neointima formation and neointimal inflammation. The results of this study will enhance understanding of the inflammatory processes of neointima formation and restenosis, in conjunction with vitamin D supplementation. Continued research is necessary to answer the question of the relationship between pro-and anti-inflammatory cytokines, IL-37, and vitamin D status in the pathogenesis of atherosclerosis, neointima formation, and restenosis [3]. The findings of this study should direct further research into the implications that IL-37 has with vitamin D and its role in neointima formation and restenosis. The correlation between IL-37 expression, neointima formation, restenosis, and vitamin D status may prove useful, not only in a deeper understanding of the underlying mechanisms but also as a diagnostic tool (or in developing targeted pharmacologic therapies for atherosclerosis, neointimal hyperplasia, and restenosis).

## 2. Results

### 2.1. Vitamin D Supplementation Attenuates Neointima Formation in Coronary Arteries after Balloon Angioplasty and Stent Placement

Hematoxylin and eosin staining of the angioplasty and stented coronary arteries showed significantly reduced neointima formation in the vitamin D supplemented (VDSupp) group, compared to the vitamin D sufficient (VDSuff) and vitamin D deficient (VDDef) groups (Figure 1A,D,G,J) in microswine with angioplasty. Vitamin D supplementation was also associated with reduced neointima formation in stented coronary arteries Figure 2A–F). These results support our previous findings of the association of vitamin D deficiency with neointima formation and the association of vitamin D supplementation with attenuated neointima formation [8,10].

### 2.2. Vitamin D Supplementation Attenuates IL-33 and IL-37 Expression in Neointima

Immunohistochemistry (IHC) of the coronary arteries with balloon angioplasty (Figure 1) revealed a significantly increased expression of IL-33 in VDDef swine in the neointima (Figure 1B,B’), compared to VDSuff (Figure 1E,E’) and VDSupp (Figure 1H,H’) microswine, as well as a higher expression in VDSuff microswine, compared to the VDSupp group (Figure 1E’,H’). IHC also revealed that IL-37 expression in the neointima was significantly higher in VDDef microswine (Figure 1C,C’), compared to VDSuff and VDSupp microswine. There was no IL-37 expression in VSSuff and VDSupp microswine (Figure 1F’,I’). The analysis of the IHC images showed significantly higher average numbers of IL-33-stained cells in VDDef microswine, compared to VSSuff and VDSupp microswine (Figure 1K). Similarly, the average IL-33-stained cell density per mm^2^ was higher in VDDef microswine, compared to VDSuff and VDSupp microswine (Figure 1L). Nearly similar bar height of average cell density in VDDef and VDSupp microswine is due to a very small neointimal area in VDSupp microswine. The average IL-33-stained intensity was also higher in VDDef microswine, compared to VDSuff and VDSupp microswine and in VDSuff microswine, compared to VDSupp microswine (Figure 1M). 

The expression of IL-37 was higher in VDDef microswine, compared to minimal to no expression in VDSuff and VDSupp microswine. Similarly, the average cell count, average cell density, and average stained intensity for IL-37 were higher in VDDef microswine, compared to VDSuff and VDSupp swine, which had minimal to no expression and thus, average cell count, average cell density, and average stained intensity in VDSuff and the VDSupp groups were not measurable. Overall, these findings suggest that vitamin D supplementation is associated with decreased neointima formation, IL-33 and IL-37 expression, average cell density, average cell count, and average stained intensity (Table 1). In stented coronary arteries, IHC revealed IL-33 expression in VDDef microswine only (Figure 2A) but not in VDSuff and VDSupp microswine. IL-37 expression was absent in stented coronary arteries of VDDef, VDSuff, and VDSupp microswine (Figure 2). 

### 2.3. Vitamin D Deficiency Is Associated with Caspase-1 Expression

Immunofluorescence staining for Caspase-1 (NLRP3 inflammasome) revealed minimal positive expression for Caspase-1 in coronary arteries with angioplasty of VDDef microswine. The immunofluorescence of coronary arteries, with angioplasty in VDSuff and VDSupp groups, did not reveal any immunopositivity for Caspase-1 (Figure 3).

### 2.4. Vitamin D Supplementation Associates with Decreased Pro-Inflammatory Macrophages

Immunofluorescence studies of coronary arteries with angioplasty in VDDef, VDSuff, and VDSupp microswine revealed increased immunopositivity for CD68 (pan macrophage marker), CD86 (pro-inflammatory M1 macrophages), and CD206 (anti-inflammatory M2a macrophages) in VDDef microswine (Figure 4D,G,J). The immunopositivity for CD68 and CD86 was higher than CD206 in the VDDef group. There was minimal to no immunopositivity for CD68 and CD86 in VDSuff and VDSupp microswine, but there was an increase in CD206 expression in the VDSupp group. There was no immunopositivity for CD163 (anti-inflammatory M2b macrophages) and IL-10 (anti-inflammatory M2c macrophages) in coronary arteries with angioplasty in any group (Figure 4). Similarly, immunofluorescence does not reveal immunopositivity for macrophages in stented coronary arteries (data not shown).

### 2.5. Vitamin D Attenuates the Expression of IL-33 and IL-37 in Lipopolysaccharides Treated Endothelial Cells In-Vitro

To validate our findings of attenuated expression of IL-33 with vitamin D in VDSuff and VDSupp microswine neointima, the endothelial cells (ECs), isolated from microswine arteries, were treated with lipopolysaccharides (LPS). The immunofluorescence of ECs treated with LPS showed increased expression of IL-33 (Figure 5G), however, ECs first treated with LPS and then with vitamin D showed attenuation of IL-33 expression, compared to IL-33 expression in LPS treatment alone (Figure 5, panel J vs. panel G). The baseline IL-33 expression in control and vitamin D-treated ECs were comparable (Figure 5A,D). Immunofluorescence revealed similar findings for the expression of IL-37 in ECs (Figure 6). These results are suggestive of an immunomodulatory role of vitamin D on IL-33 and IL-37 expression in ECs. It should be noted that the effect of LPS was more on the expression of IL-33, compared to IL-37 in ECs.

## 3. Discussion

Percutaneous coronary artery intervention, for balloon angioplasty and stent implantation, induces a vascular inflammatory response that results in immune cell infiltration and the increased secretion of pro-inflammatory cytokines, including IL-6, TNF-α, and IFN-γ [8,10,17]. Inflammation at the dilated vessel segment results in a cascade of inflammatory response, leading to ECs and vascular smooth muscle cells (VSMCs) dysfunction, as well as a change in VSMCs phenotype, leading to neointima formation and arterial restenosis [18,19]. Luminal narrowing, restenosis, in-stent thrombosis, and neoatherosclerosis (atherosclerosis of the coronary artery after stent placement) could present as late, or very late, complications, after deployment of drug-eluting stents and inflammation remains the core pathogenesis. Chronic inflammation within stent struts of the drug-eluting stents, persistent endothelial dysfunction, and lack of reendothelialization contribute to restenosis of the coronary artery [20,21]. Chronic inflammation, as the core pathogenic mechanism with these late complications, suggests that inflammation begets neoatherosclerosis, neointimal thickening, and restenosis; thus, attenuating inflammation should be the focus to decrease neointima formation, arterial thickening, and restenosis. In this study, we investigated the role of IL-33 and IL-37 in enduing inflammation in neointimal tissue in the coronary arteries undergoing balloon angioplasty and stent implantation. Additionally, we also investigated the effect of vitamin D status on the expression of IL-33 and IL-37, thereby inflammation, in these coronary arteries. The results of this study suggest that vitamin D deficiency associates with increased expression of IL-33, a proinflammatory cytokine of the IL-1 family, and vitamin D supplementation results in decreased expression of IL-33 and inflammation. IL-33 acts both intra- and extracellularly, to either enhance or resolve the inflammatory response, suggesting the context-dependent function of IL-33. The Treg cell-mediated protective role of IL-33 on atherosclerosis and lipid homeostasis has been discussed; however, growing evidence suggests that IL-33 induces an inflammatory response in ECs [22,23]. Since the denudation of ECs plays a crucial role in neointima formation and restenosis [24] (and the fact that inflammatory response in ECs plays an important role in neointima formation [22,23]), it is imperative to investigate the involved inflammatory mediators. A positive expression of IL-33 in neointimal tissues of vitamin D deficient, sufficient, and supplemental swine, in a corresponding decreasing intensity (in this study) in coronary arteries with angioplasty (Figure 1), suggests the crucial role of IL-33 in the pathogenesis of neointima formation. An attenuated immunopositivity for IL-33 in the neointima with vitamin D supplementation suggests the immunomodulatory role of vitamin D on IL-33. Additionally, decreased neointimal area and attenuated IL-33 with vitamin D supplementation suggests the beneficial effect of vitamin D or a probable synergistic effect of decreasing IL-33 and vitamin D [8,25]. Since the decreased expression of vitamin D receptors and vitamin D deficiency is associated with increased neointima formation, vitamin D supplementation seems to be an effective approach to attenuate inflammation, by decreasing IL-33 expression, as shown in this study, and the expression of TNF-α and IFN-γ [8]. The notion of vitamin D supplementation, to attenuate inflammation and reduce neointima formation [26,27], is also supported by the fact that vitamin D deficiency plays a crucial role in the pathogenesis of cardiovascular diseases [11] and a high prevalence of vitamin D deficiency in individuals with cardiovascular diseases [28]. The findings of decreased mean neointimal area, cell count, cell density, and stained intensity for IL-33 and IL-37-stained cells in the vitamin D sufficient and supplemented group (Table 1) supports the role of vitamin D supplementation in attenuating inflammation and neointima formation. It should be noted that vitamin D supplementation attenuates chronic inflammation and will have better therapeutic efficacy in chronic coronary artery disease conditions, compared to acute conditions.

The coronary arteries in the angioplasty group showed a positive expression of IL-37, in the vitamin D deficient group only (Figure 1). IL-37 is an anti-inflammatory cytokine, regulating the inflammatory immune response by inhibiting the expression, production, and function of pro-inflammatory cytokines, including the IL-1 family, which includes IL-33 [29]. IL-37 is secreted in the presence of inflammation to attenuate it, and the secretion of IL-37 positively correlates with inflammation. A positive expression of IL-37 in vitamin D deficient microswine coronary artery with balloon angioplasty and no expression with vitamin D supplementation supports the notion that vitamin D supplementation attenuates IL-33 expression and inflammation and, in turn, the decreased expression of IL-37. A decreased expression of IL-33 and IL-37 with vitamin D supplementation can also be attributed to the regulation of nucleotide-binding domain-like receptor protein 3 (NLRP3) inflammasome activation with vitamin D [30] and the proteolytic processing of interleukin-1 family cytokines with caspase-1 [31]. Since, vitamin D attenuates NLRP3 inflammasome activation [30], vitamin D supplementation to decrease the expression of caspase-1 and, thereby, decreased secretion of pro-inflammatory cytokines should be a potential therapeutic strategy; however, this warrants much research. Vitamin D supplementation, to downregulate caspase-1 expression, is supported by the expression of caspase-1 in vitamin D deficient microswine only but not in vitamin D sufficient and supplemented microswine in this study (Figure 3). 

Vascular inflammation is associated with the recruitment of immune cells, including macrophages [32]. The immunopositivity for CD68, CD86, and CD206 in the neointima of the coronary arteries, with angioplasty in the vitamin D deficient group, found in this study suggests the presence of recruited inflammatory cells in the neointima. Their presence also suggests the regulation of secretion of IL-33 cytokine, which is secreted by damaged endothelial cells and M2 macrophages (CD206+). The presence of CD68 positive and CD86 positive M1 macrophages suggests the presence of phagocytic inflammatory macrophages and inflammation in the neointima. Further, IL-33 promotes M1, and IL-37 promotes M2 and suppresses M1 macrophage phenotypic switch; increased expression of CD68 and CD86 positive cells in the vitamin D deficient group correlates with increased expression of IL-33 in coronary arteries with angioplasty in this study [33,34,35]. A decreased expression of CD68 and CD86 positive cells in vitamin D sufficient and supplemented group coronary arteries with angioplasty (Figure 4) further supports the hypothesis that vitamin D supplementation correlates with decreased inflammation and might be a therapeutic strategy to attenuate chronic inflammation, by decreasing IL-33 expression in the neointima and, thereby, restenosis [8,36] in chronic conditions, such as coronary artery with balloon angioplasty and coronary artery with a stent.

Immunostaining of the coronary arteries with angioplasty revealed an increased expression of IL-33 and IL-37 in the vitamin D deficient group and attenuation of expression with vitamin D supplementation. Increased expression of IL-33 and IL-37 with LPS in isolated ECs and attenuation of IL-33 and IL-37 expression with calcitriol treatment in LPS pretreated cells (Figure 5 and Figure 6) suggests the potential of calcitriol (the biologically active form of vitamin D) in attenuating the expression of IL-33 in ECs. LPS was used to induce IL-33 expression and simulating an inflammatory environment in-vitro [37]. There was an increased expression of IL-33 and IL-37 in the ECs with LPS treatment; whether it was a direct effect or paracrine effect warrants in-depth experiments. These in-vitro findings of attenuated IL-33 expression in LPS treated cells, with calcitriol, validated and confirmed the role of vitamin D supplementation to reduce inflammation and, thus, neointima, as observed in this study (Figure 1, Table 1).

Overall, the results of this study suggest the therapeutic role of vitamin D supplementation to attenuate inflammation and neointima formation after vascular intervention, as evident by the decreased IL-33 expression, macrophages, and neointimal area (Figure 7). It should be noted that IL-33 expression was positive in only the vitamin D deficient group, in the stented arteries, and there was no immunopositivity for IL-33 in the vitamin D sufficient and supplemented group. Also, there was no immunopositivity for IL-37 in stented arteries. This implies a decreased expression of inflammatory markers and, thereby, of inflammation with stent implantation, compared to balloon angioplasty; however, this warrants more research, with a greater number of animals. This observation might also be a result of the stent material. This is important because neoatherosclerosis occurs with drug-eluting stents, as a late complication, and is a therapeutic challenge in clinics. Additionally, these molecular processes are associated and have a regulatory role in chronic disease and may not play any role in acute disease, such as acute myocardial infarction. Thus, immunostaining and plasma levels for other markers of chronic inflammation, including IL-6, IL-1β, TNF-α, MCP-1, CCR7/CCL7, IL-8, IL-11, IL-18, and others, should be assessed to answer these concerns and understand the angioplasty and stent-related late, and very late, complications in the chronic disease condition.

## 4. Materials and Methods

### 4.1. Porcine Model of Neointimal Hyperplasia

The experimental protocol approval was obtained from Creighton University Institutional Animal Care and Use Committee (IACUC #831). The animals were housed in the Animal Resource Facility of Creighton University, Omaha, NE, and cared for, as per National Institute of Health standards. Yucatan microswine, weighing 20–45 kgs, were purchased from Sinclair Bio-resources (Windham, MA, USA). The swine were fed a special, high cholesterol diet. The high cholesterol diet (HC) consisted of 37.2% corn (8.5% protein), 23.5% soybean meal (44% protein), 20% chocolate mix, 5% alfalfa, 4% cholesterol, 4% peanut oil, 1.5% sodium cholate, and 1% lard; with 52.8% of the kilocalories from carbohydrates and 23.1% of the kilocalories from fat (Harlan, USA). The animals were divided into three groups, namely, vitamin D deficient (VDDef), vitamin D sufficient (VDSuff), and vitamin D supplemented (VDSupp) groups. The pelleted diet was either deficient in Vitamin D (TD-150251), sufficient (TD-150250: 1.5 IU/gD, HVO, 4% Chol, NaCh) with 1500 IU/D of vitamin D3, or supplemented (TD-150252: 5 IU/gD, HVO, 4% Chol, NaCh) with 3000 IU/D of Vitamin D3. Based on our previous experience, and other studies, vitamin D3 supplementation of 1500 IU/day, for sufficient, and 3000 IU/day, for the supplemented group, was added to achieve normal (21–29 ng/m) and supplemented serum levels of vitamin D in microswine [10,38,39]. Animals were housed in the Creighton Animal Facility, under controlled conditions, 12:12-h light-dark cycle at 20–24 °C, without exposure to sunlight, and fed a controlled diet, to avoid any variation in the 25(OH) D levels, due to season or diet [10]. Venous blood from an auricular vein at baseline, before the coronary intervention and before the time of euthanasia, was collected regularly to measure the serum 25(OH)D levels. The 25(OH) D level parameters for deficient, sufficient, and supplemented group were as follows: vitamin D deficient swine was ≤20 ng/mL, vitamin D sufficient swine were 30–44 ng/mL, and vitamin D supplemented swine were >44 ng/mL. Endothelial injury and atherosclerotic vascular disease induction, by balloon angioplasty and stent expansion [36] and stent placement [40], were done in Yucatan microswine, as described previously by our group. Balloon angioplasty and stent expansion were done to simulate endothelial injury/arterial injury and not to induce myocardial infarction (MI) experimentally. Coronary artery tissues (n = 3 each for VDDef, VDSuff, and VDSupp) from angioplasty and stented arteries were collected from the microswine, terminally euthanized.

### 4.2. Tissue Harvest and Processing

The swine were sacrificed, and the heart was surgically removed. The coronary arteries were dissected, removed, and fixed in 10% formalin for 24 h at room temperature on a table shaker, set on low. Tissues were processed, embedded in paraffin, and thin sections (5 μm) were cut using a microtome (Leica, Wetzlar, Germany) and placed onto glass slides. The tissue was fixed onto the slides using a slide warmer for 1 h at 60–65 °C. Tissue sections were used, both for IHC and hematoxylin and eosin for morphological studies.

### 4.3. Immunohistochemical Studies

Carotid artery tissues were subjected to immunohistochemical staining for IL-33 and IL-37 using Horseradish Peroxidase (HRP) and 3,3′-Diaminobenzidine (DAB) peroxide substrate solution, following the standard protocol in our laboratory. Rabbit anti-IL37 (ab153889) at 1:200 dilution and rabbit anti-IL33 (sc-98659) primary antibodies at 1:50 dilution were used, and a biotinylated secondary antibody was used to stain the slides. The slides were counterstained with hematoxylin to stain the nuclei. The slides were blindly checked by the board-certified pathologist. The slides were scanned with an Olympus microscope and the average number of positively stained cells, cell density per square millimeter, and average stained intensity were calculated in three different images for all tissues in the VDDef, VDSuff, and VDSupp groups.

### 4.4. Immunofluorescence Study

Deparaffinization, rehydration, and antigen retrieval were carried out before immunostaining. Immunofluorescence staining was done as per standard protocol in our laboratory. The tissues were incubated with rabbit anti-CD86 (ab53004) for M1a, rabbit anti-CD206 (ab64693) for M2a, rabbit anti-CD163 (ab87099) for M2b, rat anti-IL-10 (JES3-19F1) for M2c, and rabbit anti-caspase 1 (ab74279) at 1:200 dilution overnight at 4 °C. This was followed by washing with PBS and incubation with Alexa Fluor 594 (red) and Alexa Fluor 488 (green) conjugated secondary antibodies (Invitrogen, Grand Island, NY, USA) at 1:1000 dilution for 1 h at room temperature. DAPI (4, 6-diamidino-2-phenylindole), with mounting medium, was used to stain nuclei. Negative controls were run by using the isotype IgG antibody for each fluorochrome. Olympus inverted fluorescent microscope (Olympus BX51) was used to scan the images at 20 and 40×, with scales of 200 and 100 μm, respectively. The images were reviewed by two independent observers, blindly. 

### 4.5. Endothelial Cell Isolation and Primary Culture

Endothelial cells (ECs) were isolated from carotid arteries, collected in endothelial culture medium (ECM) supplemented with 3% penicillin-streptomycin (PS) (3× medium), and kept at 4 °C. The arteries were washed again with 3× medium. After washing, the arteries were cleansed of the adventitia and inverted inside out in a petri dish containing ECM supplemented with 10% fetal bovine serum (FBS) and 1% PS (complete medium). The inner side of the artery was gently scraped with a cell scraper and the ECM medium containing cells were centrifuged at 300× *g* for 10 min. After discarding the medium, the pellet was dissolved in 10 mL complete ECM in a T25 flask and was incubated in a humidified chamber at 37 °C and 5% CO_2_. The cells (primary culture) were checked every 48 h and the medium was changed. After getting ECs to 80–90% confluence, a portion of ECs were frozen, and the experiments were conducted with the remaining ECs. The cells up to passage three (P3) were used for all experiments. The isolated ECs were characterized by positive staining for CD31.

### 4.6. Cell Culture, Stimulation, and Inhibition Studies

The endothelial cells (ECs), isolated from swine arteries, were cultured in endothelial culture medium (ECM) supplemented with 10% fetal bovine serum and 1% penicillin-streptomycin in a humidified incubator with 5% CO_2_. For all experiments, ECs up to passage 3 were used. The cells were cultured to 75–80% confluence in a T25 flask. For immunofluorescence studies, 8000–10,000 cells were plated in each chamber of the chamber slides overnight. The cells were treated with LPS (100 ng/mL) and vitamin D (calcitriol; 50 nM) for 24 h. For combined treatment, the cells were first treated with LPS for two hours and then with vitamin D for 24 h. After treatment, the cells were formalin (10%) fixed, treated with 0.1% Triton, and washed with PBS three times after each treatment. The cells were incubated with blocking solution for 30 min at room temperature and then with primary antibodies IL-33 and IL-37 at 1:200 dilution overnight at 4 °C. The cells were washed three times, five minutes each, with PBS. The cells were treated with secondary antibody Alexa Fluor 594 at 1:1000 dilution for 30 min at room temperature. The cells were washed with PBS and mounted with mounting media with 4, 6-diamidino-2-phenylindole (DAPI). The cells were scanned with a Leica fluorescence microscope. 

### 4.7. Statistical Analysis

The statistical significance was determined by analyzing the data using GraphPad Prism v6.0 biochemical statistical package (GraphPad Software, Inc., San Diego, CA, USA). Values of all measurements are expressed as mean ± SD of three independent experiments from each of the animals in each group (vitamin D deficient, vitamin D sufficient, and vitamin D supplemented). Statistical analysis was performed using multiple group comparisons using one-way analysis of variance (ANOVA) to determine significant differences between the two groups, with a *p*-value of <0.05 considered as significant. * *p* < 0.05, ** *p* < 0.01, *** *p* < 0.001, and **** *p* < 0.0001.

## 5. Conclusions and Prospects

The data are suggestive that vitamin D-deficiency is associated with the presence of inflammation and neointima formation, and that vitamin D supplementation reduces neointima formation, as well as the expression of the mediators of inflammation, including IL-33. Consequently, it appears that vitamin D deficiency will negatively impact the ability of the neointimal cells to respond with a robust anti-inflammatory response, and vitamin D supplementation imparts a stronger anti-inflammatory response. With the fact of the higher prevalence of hypovitaminosis D with cardiovascular diseases, vitamin D supplementation might be an adjunct therapeutic strategy to attenuate inflammation and neointima formation. Further investigation into the mechanistic aspects of the molecular mechanism underlying and the relationship of IL-33 and IL-37 with vitamin D is warranted. Furthermore, the underlying mechanisms or factors causing the differential expression of IL-33 and IL-37 in the angioplasty and stented arteries alone, as well as in the context of vitamin D, should be investigated. Additionally, the role of the inflammasome-caspase1-IL33 axis in inflammation, neointima, and restenosis in angioplasty and stented arteries warrants future research.

## Figures and Tables

**Figure 1 ijms-22-08824-f001:**
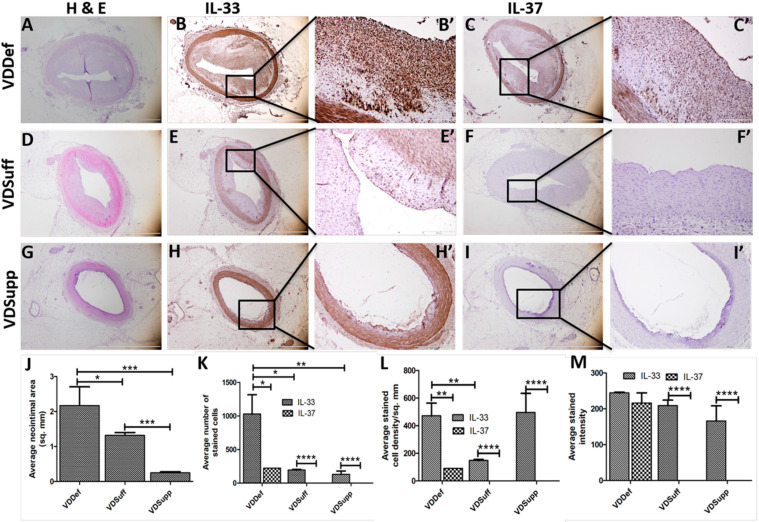
Hematoxylin and eosin and immunostaining for interleukin (IL)-33 and IL-37 in coronary arteries with angioplasty in vitamin D deficient, sufficient, and supplemented microswine. H & E staining in vitamin D deficient (panel (**A**)), sufficient (panel (**D**)), and supplemented microswine (panel (**G**)). Immunohistochemistry for IL-33 (panels (**B**,**E**,**H**)) and IL-37 (panels (**C**,**F**,**I**)) in vitamin D deficient (VDDef), sufficient (VDSuff), and supplemented (VDSupp) groups, respectively. Panels (**B’**,**C’**,**E’**,**F’**,**H’**,**I’**) are the higher magnification images for the corresponding panels. Average neointimal area (panel (**J**)), average number of IL-33- and IL-37-stained cells (panel (**K**)), average stained cell density/mm^2^ (panel (**L**)), and average stained intensity (panel (**M**)) in VDDef, VDSuff, and VDSupp microswine coronary arteries with angioplasty. All the images were scanned at 100 μm. All data has been presented as mean ± SD (n = 3). A *p*-value of <0.05 was considered significant. * *p <* 0.05, ** *p* < 0.01, *** *p* < 0.01, and **** *p* < 0.0001.

**Figure 2 ijms-22-08824-f002:**
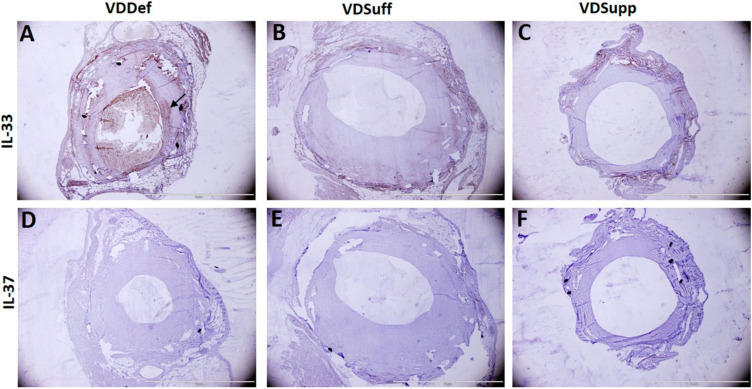
Immunostaining for interleukin (IL)-33 and IL-37 in coronary arteries with a stent in vitamin D deficient, sufficient, and supplemented microswine. IL-33 staining in vitamin D deficient (panel (**A**)), sufficient (panel (**B**)), and supplemented (panel (**C**)) coronary artery with a stent. IL-37 staining in vitamin D deficient (panel (**D**)), sufficient (panel (**E**)), and supplemented (panel (**F**)) coronary artery with a stent. All the images were scanned at 100 μm.

**Figure 3 ijms-22-08824-f003:**
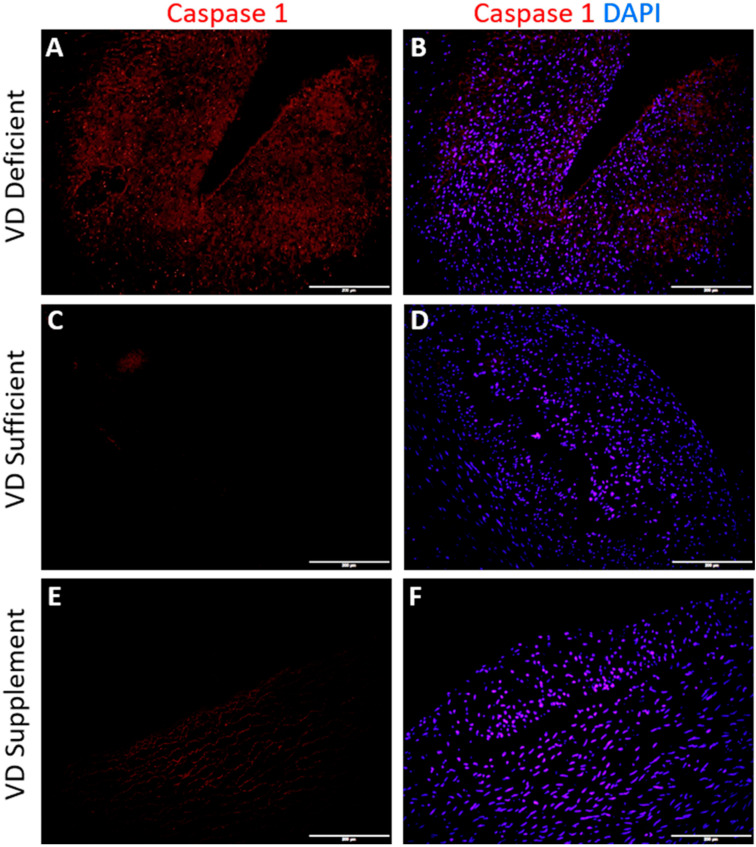
Immunofluorescence for Caspase-1 in a coronary artery with angioplasty. Caspase-1 (panels (**A**,**C**,**E**)) in vitamin D deficient (VDDef), sufficient, (VDSuff), and supplemented (VDSupp) coronary artery; caspase-1 with DAPI (panels (**B**,**D**,**F**)); 4′,6-diamidino-2-phenylindole (DAPI).All images were scanned at 200 μm scale.

**Figure 4 ijms-22-08824-f004:**
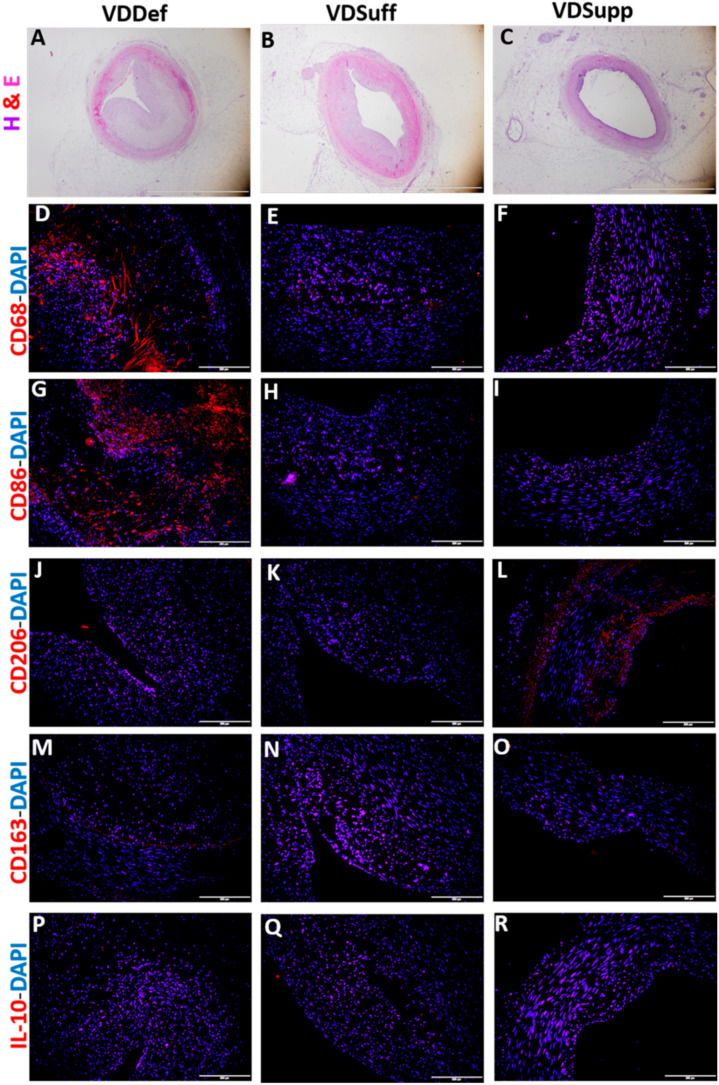
Immunofluorescence for macrophages in coronary arteries with angioplasty. Immunofluorescence staining for macrophages (CD68+ cells, panel (**D**)), M1 macrophages (CD86+ cells, panel (**G**)), M2a macrophages (CD206+ cells, panel (**J**)), M2b macrophages (CD163+ cells, panel (**M**)), M2c macrophages (IL-10+ cells, panel (**P**)); 4′,6-diamidino-2-phenylindole (DAPI, panels (**E**,**H**,**K**,**N**,**Q**)); merged images (panels (**F**,**I**,**L**,**O**,**R**)). Panels (**A**–**C**) are H & E images for the corresponding arteries in VDDef, VDSuff, and VDSupp groups. All immunofluorescence images were scanned at 200 μm and H & E images at 100 μm.

**Figure 5 ijms-22-08824-f005:**
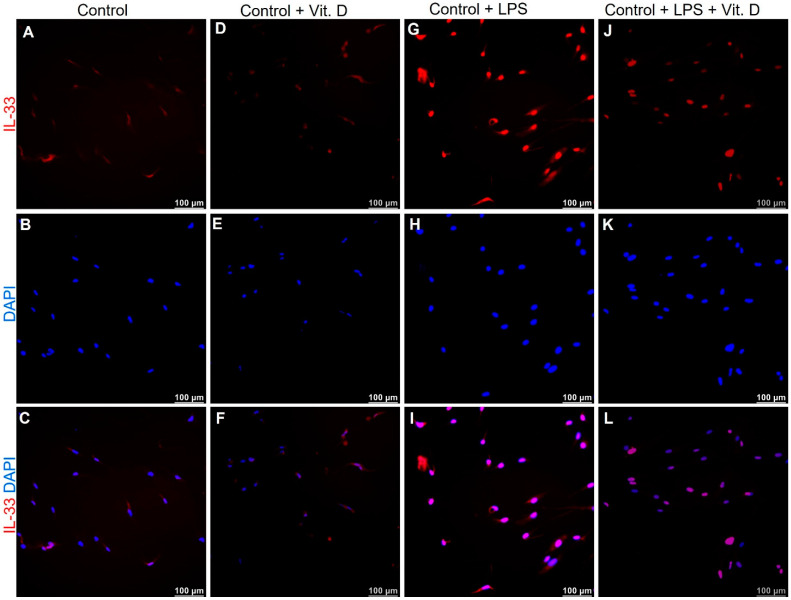
Immunofluorescence for IL-33 in endothelial cells (ECs). Immunofluorescence for IL-33 in control (panel (**A**)), calcitriol treated ECs (panel (**D**)), lipopolysaccharide (LPS) treated ECs (panel (**G**)), and calcitriol + LPS treated ECs (panel (**J**)); 4′,6-diamidino-2-phenylindole (DAPI, panels (**B**,**E**,**H**,**K**)); merged images (panels (**C**,**F**,**I**,**L**)).

**Figure 6 ijms-22-08824-f006:**
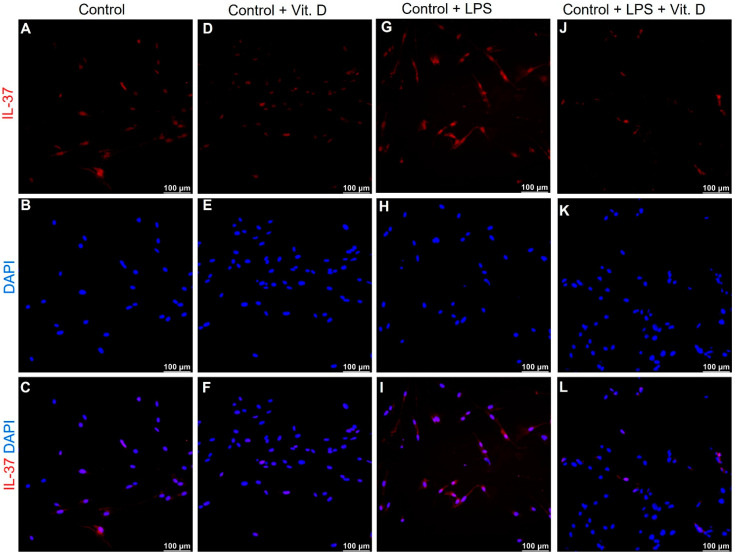
Immunofluorescence for IL-37 in endothelial cells (ECs). Immunofluorescence for IL-37 in control (panel (**A**)), calcitriol treated ECs (panel (**D**)), lipopolysaccharide (LPS) treated ECs (panel (**G**)), and calcitriol + LPS treated ECs (panel (**J**)); 4′,6-diamidino-2-phenylindole (DAPI, panels (**B**,**E**,**H**,**K**)); merged images (panels (**C**,**F**,**I**,**L**)).

**Figure 7 ijms-22-08824-f007:**
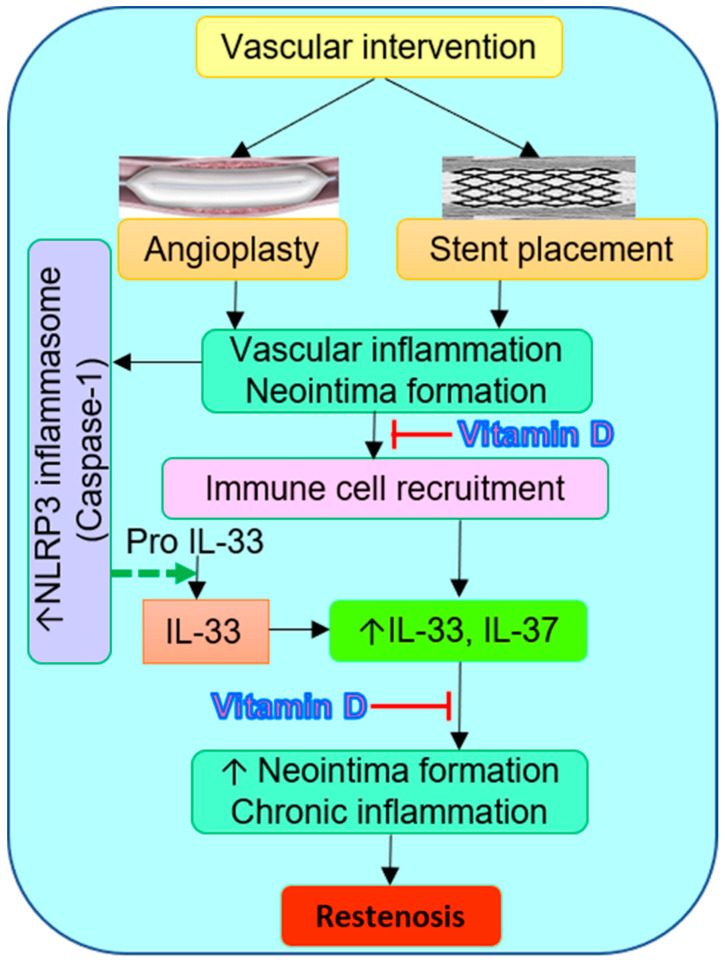
Schematic presentation of vascular intervention mediated inflammation and increased IL-33 expression, and the role of vitamin D in attenuating chronic inflammation and neointima formation and hyperplasia. Vascular intervention, namely balloon angioplasty and stent implantation, causes endothelial injury and vascular inflammatory response, leading to the recruitment of inflammatory immune cells like macrophages. This results in increased secretion of proinflammatory cytokines, including IL-33, and endues inflammation in the denuded area. Chronic inflammation and activation of various inflammatory cascades leads to neointima hyperplasia and restenosis. Vitamin D acts as an immunomodulator and anti-inflammatory agent and regulates the secretion of IL-33 and macrophages phenotypic polarization. Thus, vitamin D supplementation, to decrease the secretion of inflammatory cytokines and neointimal hyperplasia, might be a potential therapeutic strategy.

**Table 1 ijms-22-08824-t001:** Parameters measured in vitamin D deficient (VDDef), sufficient (VDSuff), and supplemented (VDSupp) Yucatan microswine. All the data are presented as mean ± S.D (n = 3).

Swine →		VDDef			VDSuff			VDSupp		
Parameter ↓		1	2	3	4	5	6	7	8	9
Mean neointimal area (mm^2^)		1.86 ± 0.08	1.43 ± 0.16	3.22 ± 0.04	1.45 ± 0.24	1.17 ± 0.04	1.34 ± 0.26	0.20 ± 0.02	0.28 ± 0.03	0.29 ± 0.05
Mean cell count	IL-33	1200 ± 81.30	472.67 ± 63.76	1419.33 ± 165.17	214.33 ± 81.88	189.33 ± 68.16	183 ± 139.37	77.33 ± 6.11	90.67 ± 15.53	226 ± 69.09
Mean cell density/mm^2^		645.16 ± 27.15	330.54 ± 78.67	441.24 ± 45.92	147.822 ± 76.90	161.36 ± 58.55	136.23 ± 119.65	393.22 ± 63.53	327.71 ± 51.46	770.45 ± 309.62
Mean stained intensity		246.86 ± 3.51	241.29 ± 5.92	245.72 ± 5.88	224.64 ± 12.55	224.31 ± 11.39	179.35 ± 55.65	128.22 ± 16.49	120.84 ± 15.25	250.23 ± 2.08
Mean cell count	IL-37	84 ± 35.55	121 ± 21.28	466.67 ± 13.32	0	0	0	0	0	0
Mean cell density/mm^2^		45.16 ± 20.53	84.62 ± 5.68	145.08 ± 2.61	0	0	0	0	0	0
Mean stained intensity		159.88 ± 14.65	250.74 ± 2.03	237.97 ± 5.70	0	0	0	0	0	0

## Data Availability

All data have been included in the manuscript.

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
