# Peer review of "Immunomodulation of IL-33 and IL-37 with Vitamin D in the Neointima of Coronary Artery: A Comparative Study between Balloon Angioplasty and Stent in Hyperlipidemic Microswine"

_ijms, 2021, doi:10.3390/ijms22168824_

Round 1

Reviewer 1 Report

The manuscript is well written, and the study is well-conducted and presented, exploring the differential expression of IL-33, IL-37, macrophages, and caspase-1 in the neointimal tissue of 18 coronary artery Yucatan microswine with vitamin D deficient, sufficient, and supplemented status. 
I would suggest refining the abstract by stating clearly the aims of this study instead of what the Authors evaluated. Then, please, present what you did to confirm your hypothesis and then your results and conclusions. 
I also suggest a moderate English grammar revision along with a double-check for typos.

Author Response

Comment: The manuscript is well written, and the study is well-conducted and presented, exploring the differential expression of IL-33, IL-37, macrophages, and caspase-1 in the neointimal tissue of 18 coronary artery Yucatan microswine with vitamin D deficient, sufficient, and supplemented status. 
I would suggest refining the abstract by stating clearly the aims of this study instead of what the Authors evaluated. Then, please, present what you did to confirm your hypothesis and then your results and conclusions. I also suggest a moderate English grammar revision along with a double-check for typos.

Response: Thank you for your encouraging comments and suggestions. We have modified the abstract as per suggestion by the reviewer and have checked the manuscript thoroughly for the English language and grammar.

Reviewer 2 Report

In this paper, Rai and Agrawal explored the role of vitamin D in the neointimal layer of the coronary artery with respect to immunomodulation of interleukins 33 (proinflammatory) and 37 (anti-inflammatory) in coronary arteries of Yucatan micro swine that were vitamin D deficient, sufficient, and supplemented with vitamin D. They found that neointimal tissue that had sufficient or supplemented vitamin D concentrations had less or no expression of IL-33 and IL-37 suggesting decreased inflammatory activity why those that had vitamin D-deficiency had upregulation of these interleukin expression pathways. Furthermore, they went on to report that attenuated inflammatory response was observed in stented arteries compared to balloon angioplasty.  They conclude that vitamin D supplementation might attenuate inflammation, neointima formation and prevent restenosis. While the findings are of interest and the manuscript is generally well-written, I suggest the following amendments to the authors, as outlined below:

  1. In the Abstract please change the word "decreased" coronary artery stenosis to less coronary artery luminal narrowing instead.
  2. Please make sure that in the Introduction you make distinction between procedural differences in terms used. For example, coronary angiography is broadly defined as a diagnostic procedure...if the coronary anatomy is then evaluated as needing intervention then the percutaneous coronary intervention (PCI) is performed, usually by stent placement or balloon dilatation, etc. So please use this terminology. In the introduction, you mention percutaneous intervention with angiography in parenthesis, angioplasty, and stent placement (which again is PCI). This is confusing from the clinical perspective. Please choose uniform terminology and apply it consistently throughout the manuscript.
  3. Authors should also acknowledge, besides restenosis and endothelial injury, that the process of so-called "neoatherosclerosis" after stent placement can also contribute to luminal narrowing and new thrombotic events. Please consider including work by Borovac et al. Eur Heart J Cardiovasc Pharmacother. 2019;5:105-116. and Niccoli et al. Int J Cardiol. 2018;258:55-58.
  4. More body of the evidence is required to be cited for the 2nd last paragraph of Introduction where there are lots of statements not substantiated with relevant references from the field.
  5. The authors declare that the research has been done with permission by Creighton University IACUC, however, none of the authors set affiliation to that institution with the manuscript. Could authors explain how is this? Both researchers are affiliated with Western University in California while they report on the study conducted at Creighton University.
  6. It should be clear in the manuscript that the authors did not experimentally induce MI but rather "simulated" arterial injury via balloon dilatation or stent expansion in the patent coronary arteries. Please make this more clear.
  7. Limitations of the study such as sample size should be emphasized. Likewise, it should be clearly noted that you performed this research in the setting of stable disease and did not perform it in the acute setting such as acute occlusion MI where the vessel is totally occluded. Therefore, this limits the conclusions and inferences of your work.
  8. This is especially important from the elective vs. emergent CAD manifestations. These mechanisms might not be that important in the acute but rather in the chronic coronary disease setting. Authors should highlight these aspects once discussing and interpreting their findings.

Author Response

Comment: In this paper, Rai and Agrawal explored the role of vitamin D in the neointimal layer of the coronary artery with respect to immunomodulation of interleukins 33 (proinflammatory) and 37 (anti-inflammatory) in coronary arteries of Yucatan micro swine that were vitamin D deficient, sufficient, and supplemented with vitamin D. They found that neointimal tissue that had sufficient or supplemented vitamin D concentrations had less or no expression of IL-33 and IL-37 suggesting decreased inflammatory activity why those that had vitamin D-deficiency had upregulation of these interleukin expression pathways. Furthermore, they went on to report that attenuated inflammatory response was observed in stented arteries compared to balloon angioplasty.  They conclude that vitamin D supplementation might attenuate inflammation, neointima formation and prevent restenosis. While the findings are of interest and the manuscript is generally well-written, I suggest the following amendments to the authors, as outlined below:

Response: Thank you for your comments.

Concern 1In the Abstract please change the word "decreased" coronary artery stenosis to less coronary artery luminal narrowing instead.

Response: Thank you for your suggestion. We have modified it in the revised manuscript.

Concern 2Please make sure that in the Introduction you make distinction between procedural differences in terms used. For example, coronary angiography is broadly defined as a diagnostic procedure...if the coronary anatomy is then evaluated as needing intervention then the percutaneous coronary intervention (PCI) is performed, usually by stent placement or balloon dilatation, etc. So please use this terminology. In the introduction, you mention percutaneous intervention with angiography in parenthesis, angioplasty, and stent placement (which again is PCI). This is confusing from the clinical perspective. Please choose uniform terminology and apply it consistently throughout the manuscript.

Response: Thank you for your comment and suggestions. We apologize for the confusion due to different terminologies used previously. We have revised the manuscript and have used the terminology “percutaneous coronary intervention’ for balloon angioplasty and stenting.

Concern 3Authors should also acknowledge, besides restenosis and endothelial injury, that the process of so-called "neoatherosclerosis" after stent placement can also contribute to luminal narrowing and new thrombotic events. Please consider including work by Borovac et al. Eur Heart J Cardiovasc Pharmacother. 2019;5:105-116. and Niccoli et al. Int J Cardiol. 2018;258:55-58.

Response: Thank you for your insightful comment and suggestion. We have included the suggested work in the revised manuscript in the discussion section (Reference # 21 and 21).

Concern 4More body of the evidence is required to be cited for the 2nd last paragraph of Introduction where there are lots of statements not substantiated with relevant references from the field.

Response: Thank you for your comment. We have included more references to support the statements in this paragraph (please see references number 11, 12, 13, 14, 15, and 16)

Concern 5The authors declare that the research has been done with permission by Creighton University IACUC, however, none of the authors set affiliation to that institution with the manuscript. Could authors explain how is this? Both researchers are affiliated with Western University in California while they report on the study conducted at Creighton University.

Response: Thank you for your comment. Dr. Devendra K. Agrawal shifted his whole lab with lab members to Western University of Health Sciences, Pomona, CA, so the current affiliation is Western University. Since the animal work was done at Creighton University, we have mentioned that we followed the IACUC protocol of Creighton University.

Concern 6It should be clear in the manuscript that the authors did not experimentally induce MI but rather "simulated" arterial injury via balloon dilatation or stent expansion in the patent coronary arteries. Please make this more clear.

Response: Thank you for your comment. We have described the process of balloon angioplasty and stent expansion more clearly in the method section under animal model subsection to make it clear that this was done only for endothelial injury and not to induce experimental MI.

Concern 7Limitations of the study such as sample size should be emphasized. Likewise, it should be clearly noted that you performed this research in the setting of stable disease and did not perform it in the acute setting such as acute occlusion MI where the vessel is totally occluded. Therefore, this limits the conclusions and inferences of your work.

Response: Thank you for your comment. We have added “Limitations of the study” section in the revised manuscript and have described the suggested limitations in the section.

Concern 8This is especially important from the elective vs. emergent CAD manifestations. These mechanisms might not be that important in the acute but rather in the chronic coronary disease setting. Authors should highlight these aspects once discussing and interpreting their findings.

Response: Thank you for your comment and suggestion. We have included the text emphasizing the application of these findings in chronic disease settings in the discussion section.

Round 2

Reviewer 1 Report

The manuscript significantly improved after revisions. I have no further comments or edits.

Reviewer 2 Report

Thank you for attending to all my comments.